# Multifaceted Functions of CH25H and 25HC to Modulate the Lipid Metabolism, Immune Responses, and Broadly Antiviral Activities

**DOI:** 10.3390/v12070727

**Published:** 2020-07-06

**Authors:** Jin Zhao, Jiaoshan Chen, Minchao Li, Musha Chen, Caijun Sun

**Affiliations:** 1School of Public Health (Shenzhen), Sun Yat-sen University, Guangzhou 514400, China; zhaoj47@mail2.sysu.edu.cn (J.Z.); chenjsh59@mail2.sysu.edu.cn (J.C.); limch7@mail2.sysu.edu.cn (M.L.); chenmsh28@mail2.sysu.edu.cn (M.C.); 2Key Laboratory of Tropical Disease Control (Sun Yat-sen University), Ministry of Education, Guangzhou 514400, China

**Keywords:** CH25H, 25-hydroxycholesterol (25HC), lipid metabolism, broadly antiviral drug, emerging infectious diseases

## Abstract

With the frequent outbreaks of emerging infectious diseases in recent years, an effective broad-spectrum antiviral drug is becoming an urgent need for global public health. Cholesterol-25-hydroxylase (CH25H) and its enzymatic products 25-hydroxycholesterol (25HC), a well-known oxysterol that regulates lipid metabolism, have been reported to play multiple functions in modulating cholesterol homeostasis, inflammation, and immune responses. CH25H and 25HC were recently identified as exerting broadly antiviral activities, including upon a variety of highly pathogenic viruses such as human immunodeficiency virus (HIV), Ebola virus (EBOV), Nipah virus (NiV), Rift Valley fever virus (RVFV), and Zika virus (ZIKV). The underlying mechanisms for its antiviral activities are being extensively investigated but have not yet been fully clarified. In this study, we summarized the current findings on how CH25H and 25HC play multiple roles to modulate cholesterol metabolism, inflammation, immunity, and antiviral infections. Overall, 25HC should be further studied as a potential therapeutic agent to control emerging infectious diseases in the future.

## 1. Introduction

Frequent outbreaks of highly pathogenic viruses are becoming a severe challenge for global public health, and developing novel strategies to control viral infectious diseases is therefore a necessary scientific issue. Increasing studies have demonstrated that the signaling pathway in response to viral infections in immune cells is closely related to cellular metabolism, and there is an emerging field of so-called immune metabolism that has highlighted the dynamic interplay between immune responses and cellular metabolism. The intricacy of immune metabolism has been extensively investigated [1,2]. For example, during viral infections, the host’s metabolism is often hijacked to meet the needs of viral replication, and marked metabolic changes ensure an optimal environment for the generation of viral offspring. Both innate and adaptive immune responses in the host are induced to fight against viral infections [3].

A variety of lipid components play important roles in physiological processes. In addition to acting as structural composition and energy stores [4], lipids also play critical roles as signaling molecules in many physiological and pathophysiological activities. Among them, cholesterol-25-hydroxylase (CH25H) can catalyze cholesterol to produce a kind of oxysterol, 25-hydroxycholesterol (25HC). 25HC and its enzyme CH25H have been shown to coordinately regulate cellular cholesterol metabolism [5]. Interestingly, recent evidence showed that the CH25H gene belongs to the family of interferon-stimulating genes (ISGs), which play key roles in inflammation, innate immunity, and subsequent adaptive immune responses through interferon signaling [6]. Further studies have shown that CH25H and 25HC inhibit a variety of highly pathogenic viruses, such as human immunodeficiency virus (HIV), Ebola virus (EBOV), Nipah virus (NiV), Rift Valley fever virus (RVFV), and Zika virus (ZIKV) [7]. However, the mechanisms of its broadly antiviral activities have not been fully clarified. In this review, we summarize the current findings on how CH25H and 25HC play multifaceted functions that are involved in cellular cholesterol metabolism and immunomodulation and direct antiviral effects. This knowledge is expected to provide insight for developing potentially therapeutic drug candidates to control emerging and re-emerging infectious diseases.

## 2. CH25H and 25HC in Regulating Cholesterol Metabolism

Cellular cholesterol homeostasis is critical for cell survival and is mainly modulated by two key transcriptional regulators, sterol regulator-binding protein (SREBP) and liver X receptor (LXR). As shown in Figure 1, CH25H and 25HC are traditionally regarded as important regulators that maintain cholesterol homeostasis by inhibiting SREBP and activating LXR.

### 2.1. CH25H and 25HC

CH25H, also known as cholesterol-25-monooxygenase, belongs to the redox enzyme family and consists of 298 and 272 amino acids in human and mouse cells, respectively. CH25H is mainly localized in the endoplasmic reticulum (ER) and Golgi apparatus and catalyzes the oxidation of cholesterol to 25HC [8]. 25HC is a kind of endogenous hydroxysterol that is involved in a variety of metabolic pathways [9]. In normal physiological status, CH25H expression in most organs is low but stable, and reports showed that the level of 25HC in human and mice plasma is approximately 2–30 ng/mL [10]. Interestingly, the CH25H level is significantly upregulated when stimulated by viral infections or Toll-like receptor (TLR) agonists, and the concentration of 25HC in plasma increased to 200 ng/mL in TLR agonist-injected mice [11].

In general, CH25H and 25HC are thought to play critical roles to maintain cholesterol homeostasis. However, there are inconsistent observations between In Vitro and In Vivo studies. For example, *CH25H-/-* mice maintain intact cholesterol metabolism compared to wild-type mice [12], and patients with abnormally elevated 25HC levels demonstrate normal levels of intoxication and bile acid [13]. One explanation is the presence of an alternative pathway that compensates CH25H-mediated cholesterol metabolism In Vivo. 25HC can also be generated by 27-hydroxylase (CYP27A1), cholesterol 24-hydroxylase (CYP46A1), and Cytochrome P450 3A4 (CYP3A4), although a small proportion [14]. These data contribute to the debate regarding the role of CH25H and 25HC in cholesterol metabolism. Alternatively, recent evidence suggested that CH25H and 25HC might also act as important regulators of inflammation, immunity, and antiviral infections.

### 2.2. 25HC and SREBP

Cholesterol biosynthesis is regulated by a SREBP, which promotes the gene expression related to the mevalonate pathway, including cholesterol biosynthesis rate-limiting enzyme HMG-CoA reductase (HMGCR) [15].

In the absence of cholesterols, SREBPs bind to SREBP cleavage activator protein (SCAP) and form SREBP-SCAP complexes on ER. SREBP-SCAP complexes are then transported to Golgi apparatus, and cleaved into mature transcription factor forms by site 1 protease (S1P) and site 2 protease (S2P) in Golgi apparatus and then promote cholesterol biosynthesis [16]. 25HC regulates cholesterol metabolism by inhibiting the activities of SREBP [17].

In the presence of excessive cholesterols, 25HC binds to membrane-spanning ER protein—Insulin-induced gene 2 (INSIG2) protein to form SREBP/INSIG2/SCAP complexes, which are retained on ER and cannot transport to Golgi apparatus, leading to dysregulation of intracellular sterol metabolism [18].

### 2.3. 25HC and LXR

LXR is a transcriptional regulator of the nuclear receptor family and plays an important role in the regulation of cholesterol metabolism [19]. 25HC is reportedly one of the LXR natural ligands and can therefore regulate cholesterol metabolism by activating LXR [20].

The LXR pathway is activated when intracellular cholesterol is abundant to initiate gene expression related to the absorption, degradation, transportation, and excretion of cholesterol. 25HC or other oxysterols induce cholesterol metabolism-related gene expression in an LXR-dependent manner, such as CH25H, cholesterol sulfotransferase-2B1b (SULT2B1b), ATP-binding cassette transporter A1 (ABCA1), and ATP-binding cassette transporter G1 (ABCG1). In addition, 25HC also induces IFN-γ expression in an LXR-dependent manner, and then IFN-γ improves CH25H expression [21], and increased CH25H subsequently promotes the production of 25HC [22]. As a result, 25HC can regulate not only lipid metabolism homeostasis, but also IFN-γ-mediated immune responses through the LXR pathway [23].

## 3. The CH25H Gene Belongs to the ISG Family

Innate immunity provides the first line of defense against viral pathogens. Innate immune cells recognize pathogen-associated pattern molecules (PAMPs), such as viral double-stranded RNA or unmethylated CpG DNA motifs, through pattern recognition receptors, such as Toll-like receptors (TLRs) and RIG-like receptors (RLRs) [24]. The interferon (IFN) signaling pathways are then rapidly activated involving multiple kinases. IFNs can inhibit viral infections by activating a unique set of IFN-stimulated genes (ISGs) [25]. To date, hundreds of ISGs have been identified, but the antiviral mechanisms for most are not yet fully described [26].

Recent evidence showed that the CH25H gene is one of the ISGs, and many viral infections can induce the upregulation of *CH25H* in most mammal cells [27,28]. For example, *CH25H* can be induced by porcine reproductive and respiratory syndrome virus (PRRSV) in Marc-145 monkey kidney cells [29]. Avian leukosis virus subgroup J (ALV-J) upregulates the expression of chicken cholesterol 25-hydroxylase (*chCH25H*) in chicken peripheral blood mononuclear cells and embryo fibroblast cell lines (DF1) [30]. The level of *CH25H* in mice bone marrow-derived macrophages (BMDMs) and dendritic cells was upregulated when stimulated with lipopolysaccharide (LPS, TLR4 agonists) and poly I:C (TLR3 agonists) [31]. Our previous data also supported that *CH25H* expression is interferon-dependent in mice and rhesus monkeys [32]. This might be because both IFN active site and interferon response elements are located upstream of the *CH25H* open reading frame in the mouse genome, and the CH25H gene is upregulated in response to IFN stimulation [4]. Further studies have shown that TLR-induced *CH25H* expression in mice BMDMs is mediated by IFNR/JAK/STAT1 signal transduction [31].

However, there is some debate on whether human CH25H gene is an ISG. One study showed that both IFN-α and IFN-γ did not induce *CH25H* expression in primary human hepatocytes [33], but another report demonstrated that *CH25H* in primary human hepatocytes was significantly induced by type I interferon [34]. The exact reason for these inconsistent data is unknown and might be related to different concentrations or varying interferon stimulation times used in those studies.

## 4. Dual Roles of CH25H and 25HC in Augmenting Pro-Inflammation and Suppressing Inflammation

Persistent inflammation and abnormal immune activation are usually accompanied with chronically viral infections, and therefore an ideal antiviral drug should modulate inflammatory responses for immune reconstitution. Based on current literature, 25HC might play a dual role to repress or augment the production of inflammatory cytokines (Figure 2).

### 4.1. Augmenting Pro-Inflammation by CH25H and 25HC

Innate immune sensors, such as STING, RIG-I/MDA5, and TLRs, have evolved to detect microbial infections by identifying PAMPs and then trigger the transcription of numerous host defense-related proteins, including pro-inflammatory cytokines [35,36]. Some studies supported that 25HC induced pro-inflammatory cytokines. For example, 25HC induced the secretion of pro-inflammatory cytokines and chemokines in monocytes/macrophages from human atherosclerotic plaque and epithelial cells, such as IL-1β, IL-6, IL-8, CCL5, and macrophage colony-stimulating factor (M-CSF) [37,38]. 25HC incubation also triggered IL-1β production in co-cultures of human monocytes and vascular smooth muscle cells [39]. In addition, 25HC was reported to play a role in foam cell formation in atherosclerosis [40]. CH25H expression was increased in alveolar macrophages and alveolar cells in patients with chronic obstructive pulmonary disease (COPD), and the concentration of 25HC and IL-8 in patients’ sputum was correlated with the number of neutrophils in the lung tissue [41]. Inflammatory cytokines in human keratinocytes were induced after 25HC incubation by promoting granzyme B release, which in turn mediated the maturation of IL-1α in cutaneous inflammatory responses [42].

The mechanism for 25HC promoting the release of inflammatory mediators, including cytokines, chemokines, and growth factors, is under investigation. Previous reports showed that 25HC-induced pro-inflammation was dependent on nuclear factor kappa-B (NF-κB) signaling, and it was observed that 25HC bound to α5β1/αvβ3 integrin as a lipid ligand to activate NF-κB signaling following nucleotide-binding oligomerization domain containing 2 (Nod2) activation [43]. 25HC-induced inflammatory signals were also related to the recruitment of transcription factor activator protein-1 (AP-1) components, FBJ osteosarcoma oncogene (FOS) and Jun proto-oncogene (JUN), and to the promoters of a subset of Toll-like receptor-responsive genes [44]. Further studies showed that 25HC-induced IL-8 secretion was calcium-dependent and associated with the ERK1/2 pathway [38]. Nevertheless, 25HC contributed to the cerebral inflammation of X-linked adrenoleukodystrophy (X-ALD) via activation of the NOD-like receptor protein 3 (NLRP3) inflammasome pathway, and the inflammasome was activated by caspase-1 following the production of key pro-inflammatory cytokines IL-1β and IL-18 [45].

### 4.2. Suppressing Inflammation by CH25H and 25HC

In contrast to the previously described observations, some studies demonstrated that 25HC played a role in suppressing inflammasome activity by inducing IFN, suppressing SREBP, antagonizing inflammasomes, and inhibiting the Akt/NF-κB signaling pathway.

Type I interferon (IFN) not only has potent antiviral activity, but also has an inhibitory effect on immunity to prevent uncontrolled inflammation, which can cause significant tissue damage in some acute viral infections [46,47] and autoimmune diseases [48,49]. A major aspect of IFN-mediated inhibition is the downregulation of inflammasome activity and IL-1β production [50,51]. The IFN-stimulated gene *CH25H* promotes the production of 25HC, and 25HC can promote the production of IFN and inhibit inflammation [21,23].

Some studies showed that 25HC inhibited IL-1β production and inflammasome activity by suppressing SREBP [52,53]. Infections with some kinds of pathogens, such as Listeria monocytogenes and mycobacteria tuberculosis, promoted 25HC production, while increased 25HC blocked the SREBP pathway activation and inflammasome activity via NLRP3 to inhibit IL-1β production, and therefore CH25H-overexpressed macrophages or mice were more susceptible to Listeria bacteria or mycobacteria tuberculosis infections [54,55]. 25HC also decreased the production of the interleukin-1 family by antagonizing the activities of NLRP- and AIM2-containing inflammasomes [56,57]. A study showed that 25HC treatment attenuated the pathological changes in LPS-induced acute lung injury in mice and reduced the overwhelming release of TLR4-mediated inflammatory cytokines by binding to myeloid differentiation protein 2 (MD-2) and subsequently suppressing the LPS-activated Akt/NF-κB signaling pathway [56]. Interestingly, 25HC’s anti-inflammation ability was also tested to treat mevalonate kinase deficiency (MKD), a hereditary auto-inflammatory disorder [58].

Therefore, 25HC seems to play opposite roles to regulate inflammation responses through a complex mechanism as reported in different studies (Figure 2). The exact reason for this mechanism is unknown, but it might be related to the various concentrations of 25HC. Our published study and other data support this hypothesis. One study showed that 25HC with nanomolar concentrations had an anti-inflammation effect, while 25HC with micromolar concentrations promoted pro-inflammation [45]. Our previous study also demonstrated that 300 ng/mL of 25HC dramatically downregulated LPS-induced inflammatory responses in primary mice splenocytes and monkey PBMCs, but this anti-inflammatory effect decreased at a higher 25HC concentration [32]. In addition to the concentration differences, we posit that the treatment time differences might be another reason, but there are no experimental data to support this hypothesis, and we will test it in our next project. Further studies are also needed to clarify how 25HC precisely modulates inflammation, especially in patients with chronic infectious diseases and autoimmune-related diseases.

## 5. Regulation of Immune Responses by CH25H and 25HC

Both innate and adaptive immune responses are important to fight against viral infections, and recent studies demonstrated that CH25H and 25HC play important roles not only in regulating cholesterol metabolism, but also in modulating innate and adaptive immune responses by regulating SREBP and LXR [59,60] (Figure 2).

### 5.1. Innate Immunity

As previously mentioned, the CH25H gene belongs to the ISG family, and therefore it is not surprising that CH25H is extensively involved in innate immunity in response to viral infections. CH25H expression increases when Toll-like receptors (TLRs) are activated, which leads to increased production of 25HC [11,22,31].

Studies have shown that 25HC acts by antagonizing SREBP processing to reduce IL-1β transcription and broadly repress IL-1-activating inflammasomes. Mouse macrophages lacking CH25H and that are unable to produce oxysterol 25HC will lead to the overproduction of IL-1 family cytokines [52].

As previously discussed in the section on 25HC and liver X receptor, LXRs are activated by 25HC as endogenous ligands. LXRs are related to the innate immune response of pathogens [61,62], and the lack of LXRα/β is highly sensitive to pathogen infections [63]. 25HC seems to elicit innate immune responses via LXR.

### 5.2. Adaptive Immunity

Adaptive immunity has a very close relationship with innate immunity and can be divided into two types: B cell-mediated humoral immune responses and T cell-mediated cellular immune responses. 25HC can affect both of these types of immune responses.

Humoral immune responses depend on the migration of highly activated B lymphocytes, and the migration of B and T lymphocytes is regulated by the G protein-coupled receptor Epstein-Barr virus-induced gene 2 (EBI2) [64]. 7α-25HC, which is produced from catalyzed 25HC by oxysterol 7α-hydroxylase (CYP7B1), is the EBI2 receptor’s lipid ligand [54]. In addition to B lymphocytes, bone marrow-derived dendritic cells and the group 3 innate lymphoid cells (ILC3s) have been reported to have 7α-25HC tropism. Therefore, as a chemokine for immune cells expressing EBI2 molecules, 25HC and its metabolite 7α-25HC are involved in the regulation of innate immunity and adaptive immune responses [65]. Further research reported that when B lymphocytes were treated with nanomolar concentrations of 25HC, they inhibited the IL-2-mediated activation of B cell differentiation and reduced the production of IgA antibodies. Nevertheless, compared to wild-type mice, there were significantly higher titers of IgA and IgG antibodies in different tissues of *CH25H-/-* mice [11]. Human patients with CYP7B1 gene deficiency cannot catalyze 25HC to 7α-25HC, and therefore their concentration of 25HC is usually 100 times that of healthy people [13]. Consistent with this observation, 25HC concentrations were increased in *CYP7B1-/-* mice, and the levels of IgA antibodies in these mice were significantly lower than in wild-type mice [66]. These data indicate that 25HC negatively regulates the secretion of IgA antibodies in B lymphocytes, suggesting the regulatory role of 25HC in the humoral immune system.

25HC also affects the transformation of CD4+ T cells from effector (IFN-γ+) to anti-inflammatory (IL-10+) phenotypes by controlling cholesterol flux. Moreover, the physiological level of 25HC in human CD4+ T cells significantly inhibits the production of IL-10 for cholesterol homeostasis by reducing c-Maf, which is the master transcriptional regulator of IL-10 cytokines [67]. Our previous studies showed that 25HC selectively suppressed pro-inflammatory CD4+ T lymphocytes secreting IL-2 and TNF-α cytokines in vaccinated mice, but had no significant immunosuppressive effects on cytotoxic CD8+ T lymphocytes or antibody-producing B lymphocytes [32]. The exact mechanism for regulating the adaptive immunity by 25HC is not fully clear, so further research should be conducted in this field.

## 6. Broadly Antiviral Infections of CH25H and 25HC through Multiple Mechanisms

25HC has been identified as a broadly antiviral compound, but the mechanism is still under investigation. Based on the literature and our understanding, we summarized the current findings regarding the possible mechanisms, including manipulation of cholesterol or other metabolite biosynthesis interfering with the virus life cycle, inhibition of viral replication by binding to viral protein targets, regulation of inflammation and immunity to control virus infections, and so on (Figure 3).

### 6.1. Inhibition of Virus Adsorption, Entry, and Release by Manipulation of Cholesterol Metabolism

Previous studies showed that 25HC could loosen cholesterol molecules in cytomembranes, thus changing the location, arrangement, and solubility of cholesterol in cytomembranes [68]. Cellular cholesterol metabolism is important in the virus life cycle, and therefore it is easy to understand the antiviral ability of 25HC partly by inhibiting cholesterol biosynthesis. For example, the adsorption, entry, assembly, budding, and release of some viruses preferentially occur in cholesterol-enriched micro-domains (“lipid rafts”) of the cell membrane, and this is especially true for enveloped viruses. As a result, 25HC inhibited the replication of various enveloped viruses, including HIV, EBOV, ZIKV, rabies virus (RABV), herpes simplex virus (HSV), vesicular stomatitis virus (VSV), murine gamma virus 68 (MHV68), and Lassa virus (LASV) [7,21,69,70].

Membrane fusion is a critical step for host-virus interactions, and several factors are involved in this step, such as the attachment and binding to the receptors on the cell surface, the spatial conformation and curvature degree of membrane phospholipid bilayers, and the arrangement and mobility of phospholipid molecules [71]. CH25H and 25HC can inhibit this step by altering the stability and integrity of cholesterol-enriched cytomembranes. 25HC might insert into the cytomembrane in an abnormal direction and then interact with the polar end of phospholipid molecules, changing the normal spatial position of phospholipid molecules, thus destroying the normal function of the cell membrane and inhibiting the entry of PRRSV [72]. Nevertheless, the change in cytomembrane composition may affect the structure and function of cell receptors, thus disrupting the interactions between the virus and host cells [73].

Research showed that 25HC inhibited the entry of PRRSV with no damage to infectious virus particles, suggesting that 25HC had effects on the cellular membrane rather than the viral envelope. Cholesterol reduction or excessive consumption inhibited the entry of PRRSV into the lipid matrix of host cytomembranes [74]. 25HC treatment also dramatically decreased rabies virus infections by inhibiting viral membrane penetration. Similar to this mechanism, a recent study showed that 25HC inhibited the membrane fusion, adsorption, and entry of ZIKV and reduced the viremia and fetal microcephaly in ZIKV-infected mice and a rhesus monkey model [75].

Oxysterol-binding proteins (OSBPs) and OSBP-related proteins (ORPs) bind and transfer sterols between cellular organelles, such as the membranes of the Golgi apparatus and ER, and 25HC can interact with OSBP and OSBP-related proteins (ORPs) [76]. RNA viruses have been shown to co-opt OSBPs and ORPs to facilitate assembly and shuttle cholesterol-enriched sites where the virus replicates [77]. 25HC prevented OSBP-mediated cholesterol shuttling to the membranous scaffolds of viral replication, which contributes to the antiviral effect of 25HC against rhinoviruses [78]. The underlying mechanism for this inhibition may be related to phosphatidylinositol-4-phosphate (PI4P), which is utilized by RNA viruses to remodel the inner cytomembrane to produce a site for RNA viral replication. For example, rhinoviruses can recruit a PI4P-rich Golgi membrane and then rely on this microenvironment to facilitate viral RNA replication. This replication process always requires OSBPs, which mediate the movement of cholesterol and PI4P between the ER and Golgi apparatus at the membrane contact site [79]. OSBP was also required for hepatitis C virus (HCV) replication and membranous web integrity [80]. Therefore, OSBP is a potential drug target against RNA viruses.

### 6.2. Inhibition of Virus Replication through Direct Interactions with Viral Component

In addition to modulating sterol biosynthesis, it was also reported that CH25H and 25HC inhibited viral infections through other multifaceted mechanisms. A recent study showed that CH25H mutants having a loss of hydroxylase activity, and thus lacking the ability to regulate cholesterol biosynthesis, still suppressed the infection and replication of HCV, implying that CH25H exhibited anti-HCV activity through an alternative hydroxylase-independent mechanism. Further research revealed that CH25H could interact with HCV non-structural protein 5A (NS5A) and suppress the formation of NS5A dimers [28]. Another study showed that 25HC significantly inhibited HCV RNA genome replication by preventing the formation of membranous web, which is the site of HCV replication [34]. In addition, 25HC could effectively inhibit the late stage of HCV infection by inhibiting the maturation of SREBPs, which is a transcription factor necessary for HCV replication [81]. 25HC also changed the status of protein prenylation in some studies, and prenylated proteins are implicated in the replication of some viruses, including HCV, hepatitis delta virus, pseudorabies virus, and respiratory syncytial virus (RSV) [82,83].

One study showed that CH25H interacted with the PRRSV nsp1α protein, which is critical for viral replication and contributes to viral pathogenicity. CH25H degraded nsp1α via the ubiquitin-proteasome pathway, and thus CH25H-mediated degradation of nsp1α affected the replication and pathogenicity of PRRSV [84]. 25HC inhibited the early stages of viral reverse transcription, thereby conferring antiviral activity for HIV infection [7]. In addition, 25HC significantly inhibited the infection of bovine parainfluenza virus type 3 (BPIV3) by preventing the synthesis of virus antigenomic RNA and genomic RNA in MDBK cells [85], implying that 25HC can effectively inhibit RNA viruses.

### 6.3. Inhibition of Virus Infection by Modulating Inflammation, Innate, and Adaptive Immunity

Viral infection and TLRs ligands induced CH25H expression by activating the TRIF adaptor and its downstream IFN-β/JAK/STAT signaling pathway [31]. This indicates that CH25H is involved in inflammation and innate immunity against microbial infections [11]. CH25H and 25HC have complicated roles in regulating antiviral immunity, including inflammatory responses, innate immunity, and adaptive immunity. For example, *CH25H* expression is upregulated during viral infections including ZIKV, HIV, VSV, HCV, and murine cytomegalovirus (MCMV), and then a series of antiviral cytokines are induced, such as IFN-γ, TNF-α, and IL-2, to inhibit viral replication [28]. 25HC also acts as an amplifier of inflammatory signaling to inhibit viral infection. It was reported that 25HC treatment significantly increased the levels of IL-1β and IL-8 expression during PRRSV infection [86].

Compared with inflammation and innate immunity, antigen-specific adaptive immune responses are more powerful to control viral infections. However, there are only a very few reports regarding how CH25H and 25HC regulate adaptive immune responses. As previously mentioned, 25HC inhibits the production of IgA antibodies. We recently assessed how 25HC affected antigen-specific T lymphocyte-mediated cellular immunity, which is important for controlling viral infections. Our data demonstrated that *CH25H* and 25HC effectively inhibited the infection of simian immunodeficiency virus (SIV) by enhancing SIV-specific IFN-γ-producing cellular responses and decreasing pro-inflammatory CD4+ T lymphocytes as SIV-infected targets [32]. A similar observation of modulating T cellular immunity was recently also found in rhesus monkeys (data not published). Further research should be conducted to clarify the underlying mechanisms.

### 6.4. Other Mechanisms of Antiviral Activity by 25HC

25HC can activate genes associated with the integrated stress response (ISR), which is attributed to altered amino acid metabolism and oxidative stress, and thus results in the suppression of viral protein translation [87]. One study showed that 25HC reduced the maturation of the G1 glycoprotein N-glycan on the Lassa virus (LASV) envelope, thus preventing this virus from adsorbing and recognizing receptors on cytomembranes and reducing the infectivity of LASV [72].

In addition to interfering with membrane fusion in enveloped viruses, studies showed that CH25H and 25HC had an expanded antiviral spectrum against non-enveloped viruses, including human reovirus, human papillomavirus-16 (HPV-16), human rotavirus (HRoV), and human rhinovirus (HRhV) [88]. A recent study found that 25HC inhibited reovirus infectious sub-viral infections neither by affecting the initial binding of viral particles to cell surface receptors nor by inhibiting sterol biosynthesis and protein prenylation, but by changing the endocytic transport patterns of reovirus particles [89]. 25HC resulted in the delayed proteolysis of outer capsids and subsequent membrane permeability, thereby reducing viral infectivity. This may suggest a potential mechanism by which 25HC inhibits the infection of non-enveloped viruses.

Of note, some viruses have evolved the antagonism of antiviral activity of CH25H and 25HC. For instance, one study showed that HSV-1 abrogated the antiviral activity of CH25H via its UL41 protein because UL41 can bind to and degrade CH25H mRNA, thereby promoting the replication of HSV-1 [90]. Another study also showed that PRRSV infection significantly downregulated the expression of *CH25H* [91]. In addition, Kaposi’s sarcoma-associated herpesvirus (HHV8) reduced the expression of *CH25H* by encoding multiple miRNAs, thus promoting viral infection [92].

## 7. Conclusions and Perspective

CH25H and 25HC are reported to exert multifaceted functions including lipid metabolism and immunomodulation, which are good examples to explain the essence of immunometabolism. More importantly, as a member of the ISG family, *CH25H* and its enzymatic product 25HC have been recently identified to have broadly antiviral activities. The underlying mechanisms are not fully clarified, but include the inhibition of virus adsorption, entry, and release by manipulating cholesterol metabolism; inhibition of virus replication through direct interactions with viral components; inhibition of virus infection by modulating innate immunity and virus-specific adaptive immunity; and the activation of the ISR, changing the endocytic transport patterns and other mechanisms. 

Highly pathogenic virus outbreaks have recently become more frequent, at least partly due to crossing the species barrier from natural wildlife to humans, including avian influenza H5N1, H7N9, SARS-CoV, MERS-CoV, ZIKV, EBOV, and the coronavirus disease 2019 (COVID-19) pandemic, which is still rapidly spreading worldwide [93,94,95]. As civilization has progressed, numerous disharmonies have occurred between humans and nature. As a result, the next outbreak of severely infectious diseases is not “if,” but is the inevitable challenge of “when, where, and what diseases,” and therefore developing broad-spectrum and safe drug candidates against emerging pathogens has become a public health priority.

Considering the broadly antiviral activities of 25HC in recent publications, this compound should be further exploited as a potential therapeutic agent in preclinical and clinical studies. Of course, many scientific issues remain to be solved. What is the exact mechanism contributing to 25HC’s broad-spectrum antiviral activity? Why does 25HC rather than other hydroxy cholesterols improve antiviral activity? How are the dual roles of 25HC in augmenting pro-inflammation and suppressing inflammation modulated through a complex mechanism under different conditions? In-depth research into the molecular mechanisms can provide new methods of screening antiviral drugs and help clinicians understand the relationship between cell metabolism, immunity, and antiviral activity. Further studies also should be conducted on the safety and pharmacokinetics of 25HC. Taken together, further research into this compound as a drug candidate for broad-spectrum antiviral infections to control emerging infectious diseases should be pursued in the future.

## Figures and Tables

**Figure 1 viruses-12-00727-f001:**
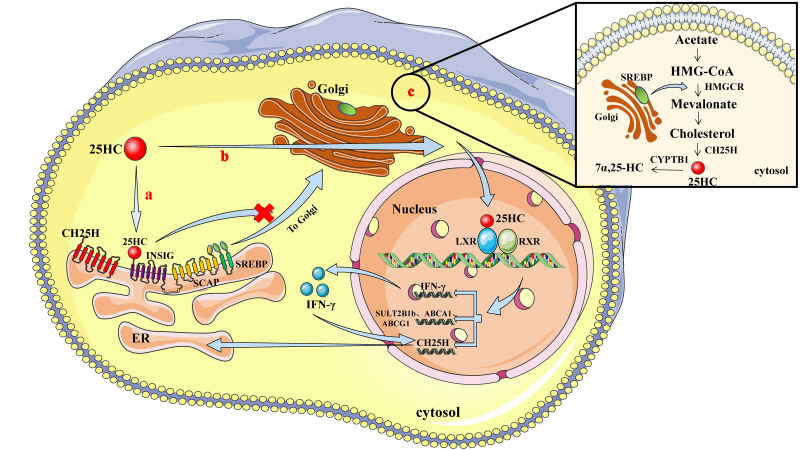
Patterns illustrating how CH25H and 25HC are involved in cholesterol metabolism. a, Regulation of sterol regulatory element-binding proteins (SREBP) by 25HC. 25HC binds to insulin-induced gene 2 (INSIG2) proteins and forms SREBP/INSIG22/cleavage activator protein (SCAP) complexes that are retained on the endoplasmic reticulum (ER) and therefore block the transportation of SREBP-SCAP complexes to the Golgi apparatus. b, As the ligand of liver X receptor (LXR), 25HC can enter the nucleus to induce the expression of cholesterol-25-hydroxylase (CH25H), cholesterol sulfotransferase-2B1b (SULT2B1b), ATP-binding cassette transporter A1 (ABCA1), ATP-binding cassette transporter G1 (ABCG1), and interferon gamma (IFN-γ), and activated IFN-γ can enhance the expression of CH25H via feedback regulation. c, SREBP promotes the expression of 3-hydroxy-3-methylglutaryl-CoA reductase (HMGCR) and subsequent cholesterol synthesis, while CH25H enzyme converts cholesterol to 25HC. 25HC 7α-hydroxylase (CYP7B1)-mediated hydroxylation contributes to converting 25HC to 7α,25-dihydroxycholesterol (7α,25HC). RXR, retinoid X receptor.

**Figure 2 viruses-12-00727-f002:**
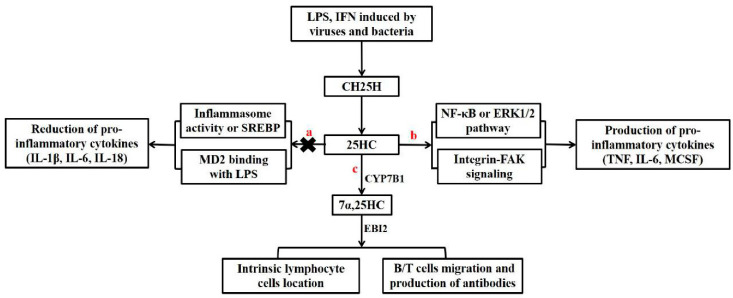
Regulation of inflammation, innate immunity, and adaptive immunity by CH25H and 25HC. The expression of CH25H, which encodes cholesterol 25-hydroxylase, and the generation of 25HC were enhanced by pathogen-induced interferon (IFN) or lipopolysaccharide (LPS). a, 25HC has anti-inflammatory effects by suppressing SREBP or inflammasome activity and interacting with myeloid differentiation protein 2 (MD2) to prevent the LPS-induced activation of Akt and the NF-κB signaling pathway. b, 25HC acts as an inflammatory amplifier to promote the production of pro-inflammatory cytokines by activating the NF-κB and ERK1/2 pathways. c, Both innate immune cells and adaptive immune cells can be regulated by 7α-25HC through its ligand-G protein coupling receptor Epstein-Barr virus-induced gene 2 (EBI2).

**Figure 3 viruses-12-00727-f003:**
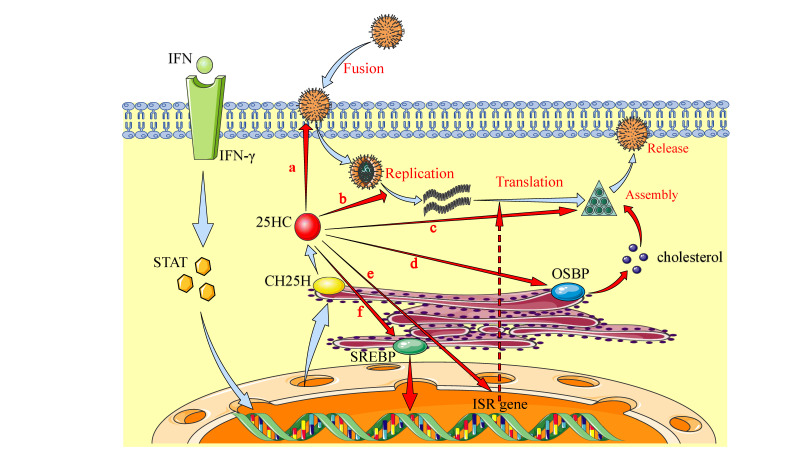
Broadly antiviral activities of 25HC through multiple mechanisms. The expression of CH25H can be induced through interferon receptor (IFNR) signaling, and then the production of 25HC is promoted from converting cholesterol. Studies have showed that CH25H and 25HC are involved in broadly antiviral activity via various mechanisms, including a, Inhibition of virus adsorption and entry by decreasing the cholesterol level of plasma membrane lipids; b, Inhibition of viral genome replication; c, Antagonizing prenylation of viral and endogenous protein that is involved in viral replication and assembly; d, Interactions with oxysterol-binding proteins to alter the cholesterol distribution; e, Activation of the gene expression associated with the integrated stress response (ISR) in macrophages, increasing oxidative stress and translation suppression; and f, Regulation of inflammation, innate immunity, and adaptive immunity.

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
