# Peer review of "Multifaceted Functions of CH25H and 25HC to Modulate the Lipid Metabolism, Immune Responses, and Broadly Antiviral Activities"

_viruses, 2020, doi:10.3390/v12070727_

Round 1

Reviewer 1 Report

The authors have sufficiently responded to my suggestions. The modifications mostly had to do with spelling, grammar, and word choice. The manuscript was not organized well but that has been improved.

Reviewer 2 Report

I am fine with this version

This manuscript is a resubmission of an earlier submission. The following is a list of the peer review reports and author responses from that submission.

Round 1

Reviewer 1 Report

The review presented by Zhao and colleagues entitled ‘Broadly antiviral activities of 25HC through modulating metabolism, inflammation and immunity’ is a comprehensive collection of literature resources on the topic. However, the title of the review implies that it is focused predominantly on the antiviral activities of the enzyme and its metabolite, but this is lost on the reader for the majority of the manuscript, until near the very end. Either the review needs to re-iterate in each section how the roles of the enzyme and its metabolites act towards restricting or altering viral infection, or the title and abstract need to be altered to reflect the actual content of the review.  Having said this, there is no comprehensive review in the literature on this enzyme and its metabolite, and a well written review would be a good addition to the field.

Major concerns

  1. This review is written very list like throughout the majority of the manuscript, making it hard to follow concepts. This review needs to be rewritten by linking pieces of list like information together, including linking sentences between concepts, providing structured paragraphs in every section that incorporate a themed idea with a good lead in sentence to orientate the reader. A review should not be a list of what is in the literature but a combination of piecing together the literature. Almost every section is difficult to follow, often with no good lead-in to random pieces of information at the start of paragraphs (there are some sections which are an exception to this statement). There are too many instances in this review to individually point out each case of presenting confused pieces of information without stating the relevance or putting it into context. However some examples are given below.
  2. Section 2 – classical function of CH25H and 25HC in regulating lipid metabolism
    1. This is a confusing section given that the authors begin by talking about the essential roles for CH25H in lipid and cholesterol metabolism, then at the end of the section discuss how knock out mice show no defects in lipid metabolism. This last statement perhaps needs to be up the front of the section, and then a discussion needs to be had during the section for the evidence both for and against its role in lipid metabolism, with commentary on the various models used, and why their might be discrepancies in the field.
    2. This section also has a small paragraph on CH25H and interferon lambda, as well as discussing its role in inflammation, which is the topic of a lower down section – it appears to be misplaced here, and confuses the section.
  3. The section discussing CH25H as an ISG is very ‘dot-point’ like in nature, listing a lot of facts, without connecting them together, which the role of a review.
  4. As with the above mentioned section, section 4. Which describes the roles of 25HC in pro-inflammation and inflammation suppression appears to be very list like in nature, without good linkage of statements and comments to each other. The last paragraph of this sections ties the information together from the above two paragraphs, but the reader is left a little lost until these statements.

Minor comments

  1. Please remove all instances of ‘etc’ throughout the manuscript and in the abstract.
  2. Line 57 – states that CH25H is mainly produced by macrophages. This is very misleading, as it can be produced by most cells, and its production in other cell types and organs is actually mentioned in other parts of this review.
  3. lines 85/86 - Nevertheless, 25HC induced the IFN-γ expression in an LXR-dependent manner, and then IFN-γ improved CH25H expression; firstly this statement needs a reference. Secondly, it is not clear what the importance of this statement is in this section.
  4. The authors go between writing CH25H and ch25h in the manuscript, please be consistent.

Reviewer 2 Report

Major issues:

  1. This paper needs to be professionally edited, because it is riddled with grammatical mistakes and spelling errors.
  2. Section 6.3 seems redundant with earlier sections discussing the role of ch25h in innate immunity. This section should be removed or combined with other discussions of innate immunity.

Minor comments:

The abstract contains abbreviations for viruses (such as EBOV). Those should be spelled out the first time they appear in the text.

It’s Zika virus (ZIKV), not “ZIKA”

“This compound” can only refer to 25HC, and not CH25H, which is a protein. Avoid using terms like “this compound” is such a general way.

Line 49: What is “direct antiviral replication?”

Line 92: The authors should include a statement about the possibility of a compensatory pathway that compensates for the absence of ch25h in vivo. That is, knockout of ch25h may not disrupt cholesterol metabolism in vivo due to the presence of an alternative pathway that compensates.

Reviewer 3 Report

In this manuscript, Zhao et al. summarized the role of Cholesterol-13 25-hydroxylase (CH25H) and 25-hydroxycholesterol(25HC) in antiviral infection. In addition, they also presented the possible molecular mechanisms underlying the antiviral activities of 25HC, postulated in recent publications. In general, this review suggests a potential of 25HC, a metabolite of cholesterol, as a novel therapeutic agent to treat viral infection, though further studies are needed. However, the manuscript was not well written and is needed to be further revised.

Major concerns:

  1. In abstract, the description of “Cholesterol-13 25-hydroxylase (CH25H) and its metabolite 25-hydroxycholesterol (25HC)” is not correct. 25HC is a metabolite of cholesterol, not CH25H that is a catalyst.
  2. 25HC has dual roles in augmenting pro- and anti-inflammation. The authors should discuss more about how the antiviral function of 25HC is through modulating inflammation. Additionally, are there any other causes for dual roles of 25HC on inflammatory response beside the concentration difference?
  3. In general, the content in the manuscript is not well organized.
  4. Editing in English is largely needed. There are many grammar and spelling errors.